# Multi-scale turbulence simulation suggesting improvement of electron heated plasma confinement

Shinya Maeyama [1✉], Tomo-Hiko Watanabe [1], Motoki Nakata[2,3], Masanori Nunami[1,2], Yuuichi Asahi[4] & Akihiro Ishizawa [5]

Turbulent transport is a key physics process for confining magnetic fusion plasma. Recent theoretical and experimental studies of existing fusion experimental devices revealed the existence of cross-scale interactions between small (electron)-scale and large (ion)-scale turbulence. Since conventional turbulent transport modelling lacks cross-scale interactions, it should be clarified whether cross-scale interactions are needed to be considered in future experiments on burning plasma, whose high electron temperature is sustained with fusion-born alpha particle heating. Here, we present supercomputer simulations showing that electron-scale turbulence in high electron temperature plasma can affect the turbulent transport of not only electrons but also fuels and ash. Electron-scale turbulence disturbs the trajectories of resonant electrons responsible for ion-scale micro-instability and suppresses large-scale turbulent fluctuations. Simultaneously, ion-scale turbulent eddies also suppress electron-scale turbulence. These results indicate a mutually exclusive nature of turbulence with disparate scales. We demonstrate the possibility of reduced heat flux via cross-scale interactions.

[1] Department of Physics, Nagoya University, Nagoya, Japan. [2] National Institute for Fusion Science, National Institutes of Natural Sciences, Toki, Japan. [3] Department of Fusion Science, The Graduate University for Advanced Studies (SOKENDAI), Toki, Japan. [4] Center for Computational Science and e-Systems, Japan Atomic Energy Agency, Kashiwa, Japan. [5] Graduate School of Energy Science, Kyoto University, Kyoto, Japan. ✉email: smaeyama@p.phys.nagoya-u.ac.jp

Plasma is an ionised gas coupled with electromagnetic fields. It is ubiquitously found in nature, laboratories, and industries, for example, in black-hole accretion discs, jets, the Sun's core, and solar wind, the Earth's magnetosphere, and magnetic fusion plasma. Magnetic fusion plasma characterised by a strong confinement magnetic field (~5 T) has steep density and temperature gradients (~10 keV/m) sustained by external heating of microwave and neutral beam injections in existing fusion experimental devices, and by fusion-born alpha particles in future burning plasma experiments. The magnetic fusion plasma is a non-equilibrium open system, self-organised with transport processes via micro-instabilities and associated turbulence driven by the steep density and temperature gradients[1]. Turbulence in magnetic fusion plasma is regarded as a multi-scale problem involving wide temporal and spatial scales, from the radius of electron gyration (~0.1 mm) to that of ion gyration (~1 cm). The electron-scale and ion-scale turbulence were often analysed separately under the scale-separation assumption. Since large-scale eddies in ion-scale turbulence were often dominant, the conventional model was designed to reproduce ion-scale turbulent transport. However, recent gyrokinetic simulation studies revealed the existence of cross-scale interactions between electron-scale and ion-scale turbulence[2–8]. Comparisons with experiments in Alcator C-Mod and DIII-D tokamaks in the United States suggest that the multi-scale interactions are necessary to explain the heat fluxes measured in the experiments[3] and play an important role in the near-future ITER device[4]. Latest campaigns in TCV, ASDEX Upgrade and JET tokamak devices in Switzerland, Germany and the United Kingdom also report that the electron-scale turbulence is responsible for a stiff dependence of electron heat flux against electron temperature gradient (ETG) in ion-heated plasma[8]. Electron-scale effects are important not only in tokamak core plasma but also in spherical tokamaks[9–11] and tokamak edge plasma[12,13]. We note that there exists another class of multi-scale problems in fusion plasma, namely, interactions between ion-scale turbulence and device scale (~1 m) or meso-scale fluctuations[14], which is beyond the scope of this work.

Beyond the existing devices, electron heating is expected to dominate in ITER. In particular for the pre-fusion power operation 1 (PFPO-1) phase, the central ratio of electron to ion temperature $T_e/T_i$ can be even larger than 3.0 by electron cyclotron heating[15]. Additionally, fusion-born alpha particles in burning plasma mainly heat electrons with keeping electron temperature $T_e$ higher than that of ions $T_i$. Due to the relaxation by ion-electron energy exchange, a reactor-relevant temperature ratio is considered around $1 < T_e/T_i < 2$. As the temperature ratio $T_e/T_i$ increases, ion-scale instabilities tend to be destabilised[16,17], in contrast to electron-scale instabilities that tend to be stabilised[18]. Therefore, the extrapolation of multi-scale interactions toward future burning fusion plasma experiments is non-trivial. To improve the performance prediction of future fusion devices, it should be clarified whether the cross-scale interactions are needed to be considered for ITER-relevant electron heated plasma and future burning plasma at $T_e/T_i > 1$.

Here, we address this problem by means of numerical simulations of multi-scale turbulence in high electron temperature tokamak plasma with electrons (e), deuterium (D) and tritium (T) fuel and helium (He) ash, mimicking future experiments on burning plasma. The turbulent transport process in magnetised plasma is well described by non-linear gyrokinetic theory[19–21], which is widely used for the analyses of magnetic fusion plasma and is also applied to accretion-disc and solar-wind turbulence[22] and auroral arc[23]. The massively parallel computation resolving from electron to ion gyroradius scales is carried out by the gyrokinetic Vlasov simulation code GKV[24,25] on the Japanese national flagship supercomputer Fugaku. The time evolution of perturbed distribution functions and electromagnetic potential fluctuations in a tokamak magnetic configuration is solved based on the delta-f electromagnetic gyrokinetic equations in a five-dimensional phase space[26]. See "Methods" for the numerical model and employed plasma parameters.

## Results

**Micro-instabilities at ion and electron scales.** In this simulation, two types of micro-instabilities can linearly grow with time because of steep electron temperature and density gradients. The first one is destabilised by toroidal precession resonance of electrons trapped in a weak magnetic field side, which is called the trapped electron mode (TEM)[27]. The other one is destabilised by the compression of the poloidal magnetic drift in torus magnetic curvature, which is named the toroidal ETG mode[28]. TEM typically possesses long wavelengths in the ion gyroradius scale and low frequencies, whereas the ETG typically has short wavelengths in the electron gyroradius scale and high frequencies. From the linear stability analysis (See Fig. 1), the poloidal wavenumber of the most unstable TEM is $k_y = 0.25\ \rho_{ti}^{-1}$ with the real frequency $\omega_r = 1.50\ v_{ti}/R_0$ and linear growth rate $\gamma = 0.145\ v_{ti}/R_0$, where $\rho_{ti}$, $v_{ti}$ and $R_0$ are respectively the ion (the hydrogen mass is used as a reference) thermal gyroradius, ion thermal speed, and tokamak major radius. The most unstable ETG mode is found at electron-scale $k_y = 4.5\ \rho_{ti}^{-1}$, $\omega_r = 24.3\ v_{ti}/R_0$ and $\gamma = 1.16\ v_{ti}/R_0$. Although the wavenumbers and frequencies of TEM and of ETG are different, the wave phase velocities in the toroidal direction are similar, that is, $(\omega_r/k_{tor})_{TEM} = 42.7\ v_{ti}\rho_{ti}/R_0$ and $(\omega_r/k_{tor})_{ETG} = 45.1\ v_{ti}\rho_{ti}/R_0$, with the toroidal wavenumber $k_{tor} = \varepsilon_r k_y/q$, where the $\varepsilon_r = r/R_0$ is the inverse aspect ratio of the torus and $q$ is the safety factor. Using an estimate of the toroidal precession drift velocity for deeply trapped electrons $v_{pre} \sim m_e v^2 q/(2erB_0)$, where $m_e$, $e$, $r$, and $B_0$ are the electron mass, electric charge, tokamak minor radius, and equilibrium magnetic field strength, the resonant condition of TEM ($v_{pre} = \omega_r/k_{tor}$) is satisfied when the particle velocity $v = 1.9\ v_{te}$. TEM has been considered relevant for experimental density scaling or electron temperature profile based on non-linear analytic theory[29,30]. Recent gyrokinetic simulation studies have also attracted researchers' attention because of the impact on the tokamak edge transport[31] and the long-standing mystery of the hydrogen

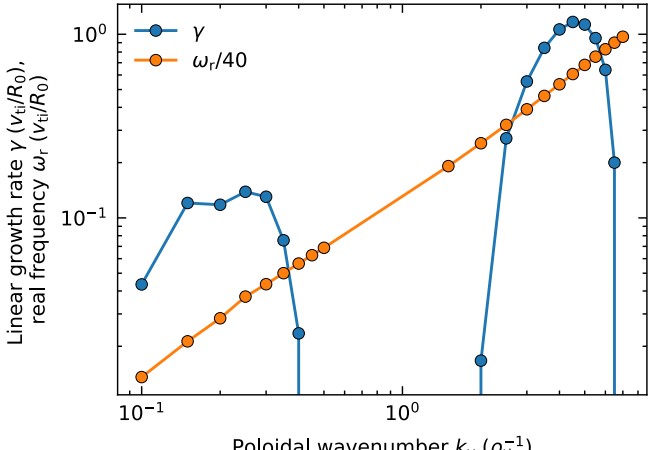

**Fig. 1 Linear dispersion relation.** Linear growth rates $\gamma$ (blue) and real frequencies $\omega_r$ (orange) in the employed plasma parameters are plotted as functions of the poloidal wavenumber $k_y$ (for $k_x = 0$). Low-wavenumber modes at $k_y < \rho_{ti}^{-1}$ are TEMs, whereas high-wavenumber modes at $k_y > \rho_{ti}^{-1}$ are the ETG modes.

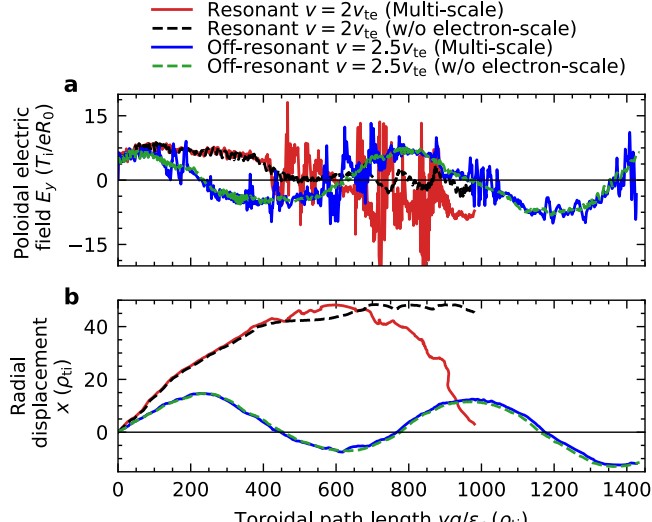

**Fig. 2 Effects of electron-scale turbulence on resonant particles.**
**a** Turbulent poloidal electric field experienced by trapped electrons, and **b** radial displacement of trapped electrons during the toroidal drift motion. Red and blue solid lines plot the trajectories for resonant ($v = 2\,v_{te}$) and off-resonant ($v = 2.5\,v_{te}$) particles, respectively, which are calculated as tracer particles under the perturbed electric field obtained by multi-scale plasma turbulence simulations of burning Fusion plasma in the time range of $162.4 < tv_{ti}/R_0 < 179.4$. For comparison, black and green dashed lines show the particle trajectories traced under the low-pass-filtered ($k_x < 4\rho_{ti}^{-1}$, $k_y < 1\rho_{ti}^{-1}$) electric field for resonant ($v = 2\,v_{te}$) and off-resonant ($v = 2.5\,v_{te}$) particles, respectively.

isotope effect[32]. Nevertheless, the cross-scale interactions between TEM and ETG have not been fully investigated yet.

**Tracer particle trajectory analysis**. The multi-scale turbulence simulations show that large-scale TEM and small-scale ETG fluctuations can co-exist in non-linearly developed turbulent flows. A typical trapped electron trajectory bounces on the weak magnetic field side of the torus outer region and makes a drift motion in the toroidal direction. The trapped electrons experience the fluctuating electric fields of TEM and ETG turbulence during the toroidal drift motion, as plotted in Fig. 2a. The poloidal electric fields of TEMs affect particles close to the resonant velocity ($v = 2\,v_{te}$, $v_{pre} = 47.3\,v_{ti}\rho_{ti}/R_0$) over a toroidal path length $yq/(\varepsilon_r\rho_{ti}) < 400$, which causes radial motion by the $E \times B$ drift (Fig. 2b). Further, an off-resonant particle ($v = 2.5\,v_{te}$, $v_{pre} = 73.9\,v_{ti}\rho_{ti}/R_0$) travels faster than waves, feels positive and negative electric fields alternately, and has no net radial displacement. Since the ETG modes also propagate with the phase velocity close to that of TEMs, small-scale ETG electric fields are also averaged out for the off-resonant particles (seen as short spikes of the blue line in Fig. 2a). However, the effects of the ETG modes on resonant particles are significant because the ETG modes also propagate along with the toroidal drift of the resonant particles. Further, the trajectory of a resonant particle is disturbed by small-scale turbulence (Fig. 2b). A secular displacement is observed (the red line in Fig. 2b after $yq/(\varepsilon_r\rho_{ti}) > 700$) because the tracer particle trajectory is modified by small-scale turbulence and moved in an oppositely rotating large-scale eddy. Since the statistical correlation between turbulent flows and perturbations of plasma distribution functions determines the resultant turbulent transport, the impacts of small electron-scale turbulence on turbulent transport are quantitatively investigated in the following analyses.

**Turbulent fluctuation profiles**. Perturbed electron pressure and the streamlines of turbulent $E \times B$ flows are plotted in Fig. 3a. Radially elongated fluctuations of TEMs are observed in the torus outer region, so-called the bad curvature region. Streamlines of the $E \times B$ flows are superimposed on the colour map, namely, turbulent $E \times B$ flows cause electron thermal transport. According to the flow directions, high-temperature fluctuations go outward and low-temperature fluctuations inward. Velocity–space dependence of turbulent flux in Fig. 3c shows that trapped particles (surrounded by green lines) satisfying the precession drift resonance condition (on black dashed line) are responsible for the turbulent thermal transport. In the magnified picture of perturbed electron pressure and streamlines in Fig. 3b, small-scale ETG turbulent eddies are observed to co-exist with large-scale TEM fluctuations. This means that the small-scale ETG turbulence disturbs the streamline of $E \times B$ flows, modifies the correlation between turbulent flows and perturbed electron pressure, and possibly affects electron heat transport.

**Turbulent transport spectra**. To examine the effect of ETG turbulence on TEM-driven turbulent transport, multi-scale simulation results are compared with an ion-scale simulation resolving only TEM scales and an electron-scale simulation resolving only ETG scales (for details, see "Methods"). The resultant poloidal wave number ($k_y$) spectra of electron energy flux are shown in Fig. 4. The multi-scale spectrum shows two peaks attributed to low-$k_y$ TEM and high-$k_y$ ETG turbulence. Low-$k_y$ TEM components in the multi-scale simulation are reduced compared with those in the single ion-scale simulation, suggesting the suppression of TEM by ETG turbulence. The cross-scale interactions modify ion-scale turbulent fluctuation amplitude, which affects turbulent transport levels of not only electrons but also fuel D, T, and He ash. The turbulent heat fluxes in multi-scale TEM/ETG turbulence [$Q_e$(TEM/ETG) = 5.66 $Q_{gB}$, $Q_D + Q_T$(TEM/ETG) = 0.24 $Q_{gB}$, $Q_{He}$(TEM/ETG) = 0.02 $Q_{gB}$] are reduced compared with those in single-scale TEM turbulence [$Q_e$(TEM) = 33.63 $Q_{gB}$, $Q_D + Q_T$(TEM) = 1.11 $Q_{gB}$, $Q_{He}$(TEM) = 0.09 $Q_{gB}$] because of the suppression of TEM turbulence by ETG turbulence, where $Q_{gB} = n_e T_e c_a \rho_a^2/R_0^2$ is the gyro-Bohm unit with the speed of acoustic wave $c_a = \sqrt{(T_e/m_i)}$ and the acoustic gyroradius $\rho_a$. The agreement of high-$k_y$ ETG peaks of turbulent flux spectra in multi-scale and single electron-scale simulations seems coincident. Although ETG turbulence initially saturates at a higher transport level, ETG-driven zonal flows suppress turbulent transport after a long time $tv_{ti}/R_0 \geq 50$ in the single electron-scale simulation (not shown), which is consistent with a recent study on single-scale ETG turbulence[10]. Therefore, the suppression of ETG turbulence by ETG-driven zonal flows dominates the single electron-scale simulation result, whereas no significant zonal flow is observed in the multi-scale simulation, indicating the existence of suppression mechanisms of ETG turbulence by cross-scale interactions.

**Gyrokinetic non-linear entropy transfer analysis**. The cross-scale interactions between the TEM and ETG modes can be interpreted in two ways. Moving from the large to small scales, the suppression of ETG turbulence in the appearance of TEMs is due to the distortion of the electron-scale streamers by the ion-scale turbulent eddies, as observed in previous multi-scale simulations[2,33]. In contrast to a previous work discussing the suppression of ETG modes by TEM-driven zonal flows (poloidally symmetric flows)[34], in this study, there are no strong zonal flows (Fig. 3a). Our results indicate that the shearing of TEM turbulent eddies (not necessarily zonal flows) can suppress ETG turbulence. Moving from small to large scales, the reduction of TEM turbulent transport in the

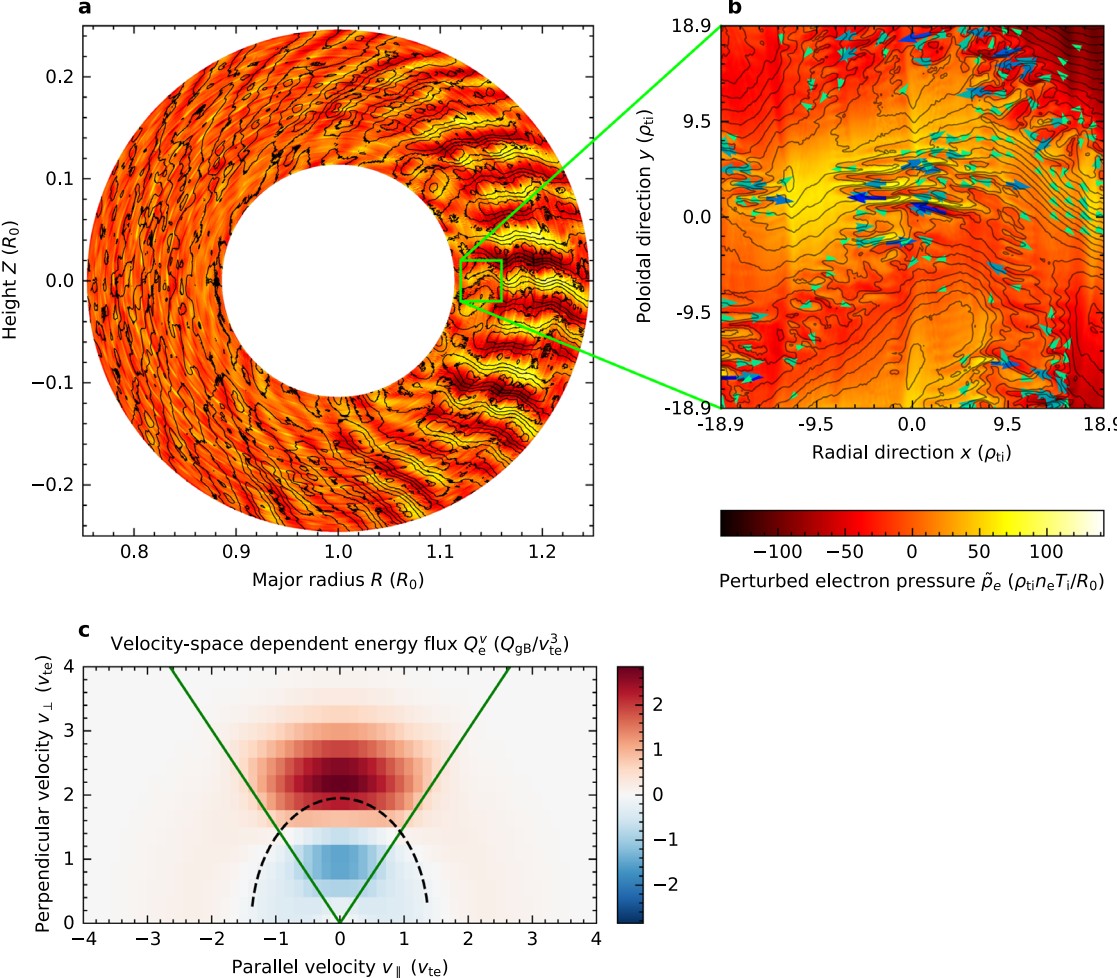

**Fig. 3 Mechanisms of electron heat transport in multi-scale turbulence. a** Poloidal cross-section of perturbed electron pressure $p_e$ (colour contour) and streamlines of $E \times B$ drift flows (solid black lines). **b** Magnified picture showing $E \times B$ drift velocity by arrows. **c** Velocity–space-dependent turbulent energy flux of electrons $Q_e^v$ at the poloidal angle $\theta = 0$, averaged over time $100 < tv_{ti}/R_0 < 200$. The green solid and black dashed lines show trapped-passing boundaries and the precession drift resonance condition, respectively, for the TEM phase velocity $(\omega_r/k_{tor})_{TEM} = 42.7\, v_{ti}\rho_{ti}/R_0$.

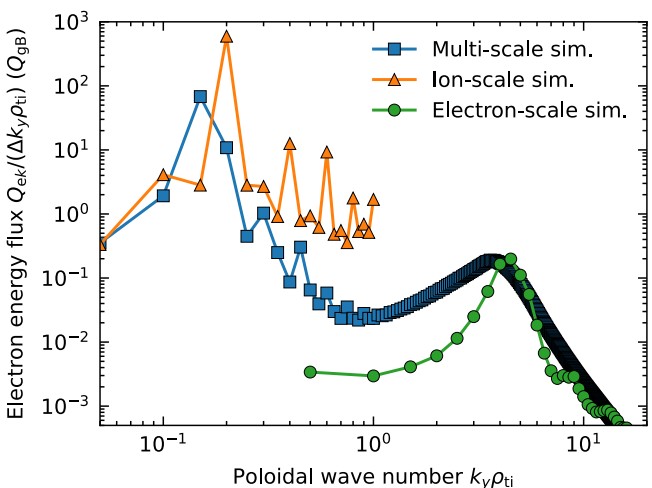

**Fig. 4 Poloidal wavenumber spectra of the time-averaged electron energy flux $Q_e$.** The multi-scale TEM/ETG turbulence simulation result is plotted using a blue line. Results of a single ion-scale TEM turbulence simulation ($k_y\rho_{ti} \leq 1$) and single electron-scale ETG turbulence simulation ($k_y\rho_{ti} \geq 0.5$) are plotted using orange and green lines, respectively.

presence of ETG turbulence is in contrast to the multi-scale simulations of ion-temperature-gradient (ITG) modes[2] but resembles the suppression of micro-tearing modes (MTM) by ETG turbulence[5]. The non-linear cross-scale interactions between electron-scale ETG turbulence and ion-scale TEM turbulence are investigated using the gyrokinetic non-linear entropy transfer analysis[35]. The perturbed entropy is a measure of amplitude fluctuations of the plasma distribution function, and its non-linear excitation or damping is described by the entropy transfer function. The net entropy transfer to a mode with wave number **k** is split into contributions of the electron-scale, electron-scale coupling $J_k^{\Omega e,\Omega e}$ and the ion-scale, electron-scale coupling $2J_k^{\Omega i,\Omega e}$ using the subspace transfer analysis technique[36] and defining the ion and electron scales $\Omega_i$, and $\Omega_e$ in the perpendicular wavenumber space. Spectra of the time-averaged transfer function in Fig. 5a shows that the electron-scale, electron-scale coupling has a negative contribution on low-$k_y$ TEM fluctuations (Fig. 5a, $J_k^{\Omega e,\Omega e} < 0$ at $k_y\rho_{ti} < 0.5$), which directly confirms the damping effect of ETG turbulence on TEMs. From the energy conservation relation among triad subspaces called the detailed balance, $J_{k\in\Omega i}^{\Omega e,\Omega e} + 2\,J_{k\in\Omega e}^{\Omega i,\Omega e} = 0$, the suppression of TEMs by ETG turbulence ($J_k^{\Omega e,\Omega e} < 0$ at $k_y\rho_{ti} < 0.5$) indicates entropy transfer from low-$k_y$ TEMs to high-$k_y$ modes ($2J_{k\in\Omega e}^{\Omega i,\Omega e} = -J_{k\in\Omega i}^{\Omega e,\Omega e} > 0$). As plotted in Fig. 5b, net entropy gain in high-$k_y$ range via ion-scale coupling is observed at finite $k_x$ ($J_k^{\Omega i,\Omega e} > 0$ at $k_y\rho_{ti} \sim 4$ and $k_x\rho_{ti} > 1$) but not at ETG peaks ($k_y\rho_{ti} \sim 4$ and $k_x\rho_{ti} \sim 0$, characterised

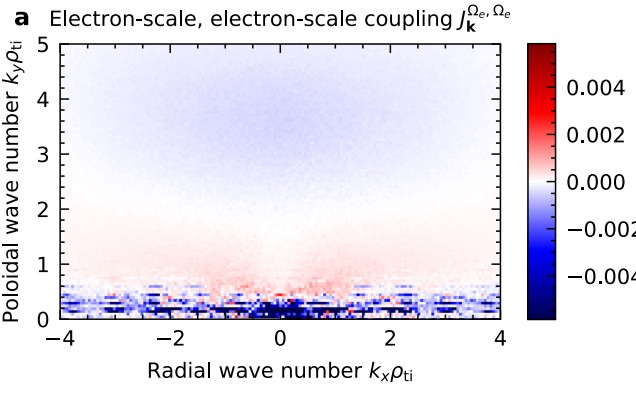

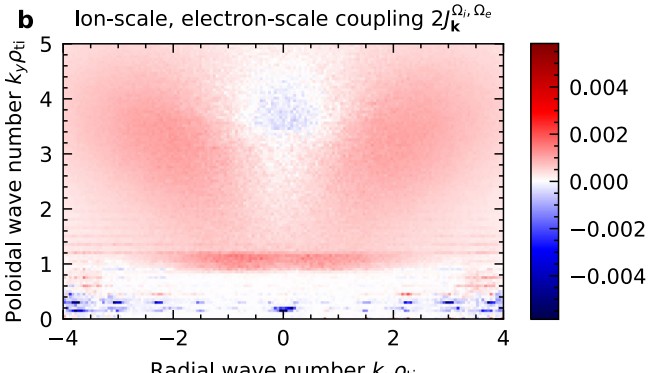

**Fig. 5 Perpendicular wavenumber spectra of the time-averaged non-linear electron entropy transfer in the multi-scale turbulence simulation.** **a** The contribution of the electron-scale, electron-scale coupling $J_{\boldsymbol{k}}^{\Omega e,\Omega e}$. **b** The contribution of the ion-scale, electron-scale coupling $2J_{\boldsymbol{k}}^{\Omega i,\Omega e} = J_{\boldsymbol{k}}^{\Omega i,\Omega e} + J_{\boldsymbol{k}}^{\Omega e,\Omega i}$, as normalised by $n_e T_i v_{ti}\rho_{ti}^2/R_0^3$.

by a peak of energy specrum at high wavenumber, which is close to linearly unstable ETG modes). These high-$k_\perp$ modes create higher $k_\perp$ fluctuations ($J_{\boldsymbol{k}}^{\Omega e,\Omega e} > 0$ at higher $k_y\rho_{ti} > 7$, not shown), which are eventually damped by collisional dissipation. Additionally, the ion-scale, electron-scale coupling has a negative contribution on ETG ($J_{\boldsymbol{k}}^{\Omega i,\Omega e} < 0$ at $k_y\rho_{ti} \sim 4$ and $k_x\rho_{ti} \sim 0$), confirming the suppression of ETG modes by ion-scale turbulence.

**Extrapolation of multi-scale turbulent interactions toward high $T_e/T_i$ regime.** Finally, the dependence of turbulent energy flux $Q_e$ on the temperature ratio $T_e/T_i$ is examined in Fig. 6. Corresponding linear dispersion relations (as in Fig. 1) and poloidal wavenumber spectra of electron energy flux (as in Fig. 4) for each $T_e/T_i$ are respectively shown in Supplementary Figs. 1 and 2. Since ETGs are highly unstable at $T_e/T_i = 1$, turbulent transport in the multi-scale simulation agrees with that in the single electron-scale simulation. As the electron temperature increases, ETGs are stabilised (green line), whereas TEMs are destabilised (orange line). At $T_e/T_i = 4$, TEM dominates turbulent transport even in the multi-scale simulation. The $T_e/T_i = 3$ case is thoroughly investigated in this study, where TEMs are close to marginal stability and significantly affected via the cross-scale interactions with ETG turbulence. From the ITER profile modelling and a reference value of a high $T_e$ discharge of the DIII-D tokamak[37], a reactor-relevant temperature ratio is considered around $1 < T_e/T_i < 2$ due to the ion–electron energy exchange. Our survey of dependence of $Q_e$ on $T_e/T_i$ covers this range. An important consequence is that the ETG contribution survives in a wider parameter range $1 < T_e/T_i < 3$, although it may have been considered that ETG could have non-negligible contribution when $T_e \sim T_i$ and sufficiently large electron temperature[8]. Additionally, the multi-scale turbulence simulation

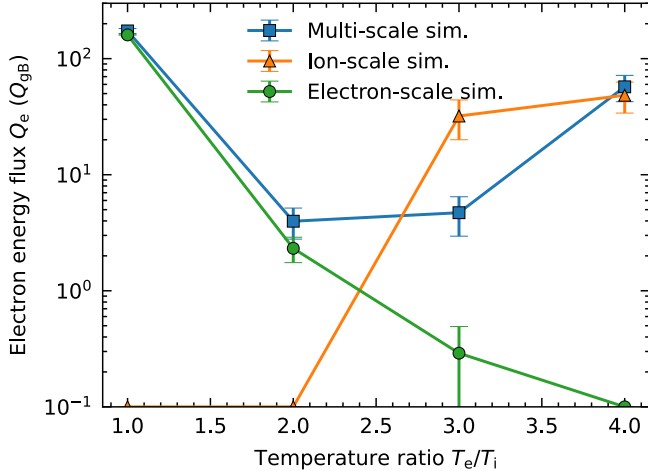

**Fig. 6 Electron energy flux $Q_e$ as a function of the temperature ratio $T_e/T_i$.** The multi-scale TEM/ETG turbulence simulation result is plotted using a blue line with standard deviation error bars. The results of a single ion-scale TEM turbulence simulation ($k_y\rho_{ti} \lesssim 1$) and single electron-scale ETG turbulence simulation ($k_y\rho_{ti} \gtrsim 0.5$) are plotted using orange and green lines, respectively. Note that an electron-scale simulation at $T_e/T_i = 4$ and ion-scale simulations at $T_e/T_i = 1$ and 2 contains no unstable modes (see Supplementary Fig. 1) and decays in time, and therefore cannot define finite amplitudes. The corresponding points are plotted on x-axis. A large standard deviation bar for the case of the electron-scale simulation at $T_e/T_i = 1$ (268 $Q_{gB}$) is omitted for visibility.

shows the existence of an appropriate $T_e/T_i$ range where the cross-scale interactions suppress turbulent transport. The suppression of near-marginal TEM by ETG turbulence leads the upshift of the critical temperature ratio for an increase of TEM-dominated energy flux (from $T_e/T_i \sim 2$ in the ion-scale simulation to $T_e/T_i \sim 3$ in the multi-scale simulation), analogous to the Dimits upshift of critical ion temperature gradient where zonal flows suppress near marginal ITG modes[38].

**Discussion**
A series of our works on multi-scale turbulence in magnetised plasma indicates some common features: large-scale turbulence tends to suppress small-scale turbulence (ITG/ETG[2–4], MTM/ETG[13], and TEM/ETG in this study), whereas small-scale turbulence tends to destroy large-scale structures (damping of short-wave-length zonal flows by ETG[36], destruction of radially localised current sheet of MTM by ETG[5] and disturbance of drift resonance between trapped electrons and TEM by ETG in this study). These findings suggest the mutual conjunction between disparate-scale turbulence as a generic nature of cross-scale interactions beyond a conventional turbulence theory described using single-scale energy injection/cascade/dissipation processes.

Our results answer the question of whether the cross-scale interactions are needed to be taken into account for future burning plasma experiments. Even beyond the existing ion-heated devices $T_e \sim T_i$, electron-scale turbulence can have impacts in electron heated plasma with high electron temperature $T_e > T_i$. This study demonstrates the possibility of reducing total electron heat flux via cross-scale interactions. Considering the opposite dependence of TEM and ETG instabilities on the temperature ratio $T_e/T_i$, our results reveal the existence of an appropriate $T_e/T_i$ range where the cross-scale interactions reduce turbulent transport, which will be advantageous for optimum tokamak operation. This finding is particularly important in burning plasma, because electron heating by the fusion-born alpha particles keeps electron temperature high $T_e > T_i$. It also has impacts on

understanding electron-heated plasma at the PFPO-1 phase in the ITER research plan, contributing the early success of fusion energy development.

This work has extensively investigated the cross-scale interactions between TEM and ETG turbulence by assuming electron temperature and its gradients exceed those of ions, though the dominant instabilities at ion scale (e.g, ITG, TEM, and MTM) depend on magnetic configuration and plasma parameters. This work is also limited to simulations with electrons, deuterium and tritium fuels, and helium ash, excluding energetic alpha particle dynamics. It has been reported that TEM drives low-level transport of alpha particles[39]. The resonant interactions between energetic particles and micro-instability will be significant for ITG modes[40,41] but not for TEM and ETG modes because the propagation directions are opposite[42]. Therefore, interactions between energetic particles and TEM/ETG turbulence are expected to be weak. When we consider interplay from electron to ion-scale turbulence and macroscopic fluctuations, magnetohydrodynamic instability driven by energetic particles will become a candidate for the third player[43], which is a theoretically and numerically challenging subject.

## Methods

**Simulation details**. Micro-instabilities and turbulent transport in magnetised plasma are simulated using the gyrokinetic Vlasov simulation code GKV[24,25]. The code is parallelised by a hybrid of message-passing interface and open multi-processing. Optimisation techniques for high-performance computing are implemented, such as the pipelined computation-communication overlap, segmented process mapping on the three-dimensional torus interconnect[25], and communication-avoiding iterative solver for the implicit collision operator[44]. This implementation ensures that the GKV code achieves a good scalability up to 12,288 nodes on the Fugaku supercomputer, 3.1 Peta-FLOPs (floating-point operation per second), which comprises 7.5% of the theoretical peak performance of the architecture and parallel efficiency of 83.7%. The highly optimised code and plentiful computational resources of Fugaku enable us to investigate the unexplored multi-scale nature of turbulent transport in burning fusion plasma.

Time evolution of the perturbed distribution functions $f_s$ of plasma species s, the electrostatic potential $\phi$, and the magnetic vector potential parallel to the equilibrium magnetic field $A_\parallel$ is solved based on the delta-$f$ gyrokinetic Vlasov–Poisson–Ampère equations[26]. The configuration space is represented by field-aligned coordinates $x = r − r_0$, $y = [q(r)\theta − \zeta]r_0/q(r_0)$, $z = \theta$, where $r$, $\theta$, and $\zeta$ are the tokamak minor radius, and the poloidal and toroidal angles, respectively. The velocity–space coordinates are the parallel velocity $v_\parallel$ and magnetic moment $\mu$. The local flux-tube model[45] resolves perturbed quantities in a long and thin simulation box along a field line, whereas the equilibrium quantities are approximated at the flux-tube centre $r = r_0$, consistent with delta-$f$ gyrokinetic ordering. Then, periodicities in perpendicular directions $x$ and $y$ are assumed along with homogeneous turbulence in fluids. The last boundary condition is the torus periodicity $f(r, \theta + 2\pi, \zeta, v_\parallel, \mu) = f(r, \theta, \zeta, v_\parallel, \mu)$. In this study, the equilibrium magnetic field is assumed to have a concentric circular torus geometry (the so-called s-α model with geometric $\alpha = 0$), which is characterised by the tokamak inverse aspect ratio $\varepsilon_r$, safety factor $q$, and magnetic shear $\hat{s}$. We selected $\varepsilon_r = 0.18$, $q = 1.42$ and $\hat{s} = 0.8$. The following plasma species were included in simulations: electron, deuterium, tritium, and helium (s = e, D, T, and He). The employed plasma parameters are $e = e_D = e_T = −e_e = e_{He}/2$, $m_i = 1837\ m_e = m_D/2 = m_T/3 = m_{He}/4$, $T_i = T_D = T_T = T_{He}$, $n_D = n_T = 0.45\ n_e$, $n_{He} = 0.05\ n_e$, $R_0/L_{ne} = R_0/L_{nD} = R_0/L_{nT} = R_0/L_{nHe} = 3$, $R_0/L_{Te} = 9.342$ and $R_0/L_{TD} = R_0/L_{TT} = R_0/L_{THe} = 1$, where $L_{ns} = −den\ n_s/dx$ and $L_{Ts} = −den\ T_s/dx$ are the density and temperature gradient scale lengths. The elementary electric charge $e$, hydrogen mass $m_i$, and tokamak major radius $R_0$ are used as references. The charge density satisfies the quasi-neutrality condition $\sum_s e_s n_s = 0$ and $\sum_s e_s n_s/L_{ns} = 0$. The plasma beta value is $\beta = \mu_0 n_e T_D/B_0^2 = 0.05\%$, normalised Debye length is $\lambda^2 = \varepsilon_0 B_0^2/(m_i n_e) = 10^{-3}$ and normalised collision frequency is $\nu^* = qR_0\tau_{ee}^{-1}/(\sqrt{2}\varepsilon_r^{3/2}v_{te}) = 0.05$ with electron–electron collision time $\tau_{ee}$. The above plasma parameters are chosen from a previous study[32] but the number of plasma species is increased. Most parameters are comparable to the existing tokamak device (e.g., a high $T_e$ discharge of DIII-D #173147[37]), whereas the electron temperature and its gradient are slightly enhanced to mimic the electron-heated burning fusion plasma. Although electron heating by the alpha particles is the main heat source in the burning plasma, the ion–electron energy exchange process also heats ion species. When the ITG increases due to the energy exchange, the ITG modes are destabilised. For the multi-scale ITG/ETG turbulence at $T_e = T_i$ and $R_0/L_{Te} = R_0/L_{Ti}$, detailed mechanisms of cross-scale interactions between ITG and ETG modes have been reported previously[2,36]. In this study, we focus on the parameter regime at $T_e > T_i$ and $R_0/L_{Te} > R_0/L_{Ti}$ which has not yet been analysed in existing devices. We

examined the electron temperature dependence in the range $1 \le T_e/T_i \le 4$, and detailed analyses are presented for the case of $T_e/T_i = 3$, where ETG turbulence significantly affects near-marginal TEMs. These parameters are also related to the electron-heated plasma at the PFPO-1 phase in the ITER research plan which is an important step for early success of ITER. The prediction of the integrated modelling of PFPO-1 phase $H$-mode plasma reported high central temperature ratio $T_e/T_i > 3$ and $R_0/L_{Te} > R_0/L_{Ti}$[15]. For the multi-scale turbulence simulations, we employed simulation box sizes $0 \le x/\rho_{ti} < 125$, $0 \le y/\rho_{ti} < 40\pi$, $-\pi \le z < \pi$, $-4.5 \le v_\parallel/v_{ts} \le 4.5$, $0 \le \mu B_0/T_s \le 12.5$, and grid points in each dimension $(N_x, N_y, N_z, N_{v\parallel}, N_\mu) = (2048, 2048, 40, 64, 16)$ for the $T_e/T_i = 1$ case and $(N_x, N_y, N_z, N_{v\parallel}, N_\mu) = (1024, 1024, 40, 64, 16)$ for the $2 \le T_e/T_i$ cases. Since the GKV code treats perpendicular $x$ and $y$ space using the Fourier spectral method with 2/3 de-aliasing rule, $f(x, y, z, v_\parallel, \mu) = \sum_{kx}\sum_{ky} f_k(z, v_\parallel, \mu)\exp(ik_x x + ik_y y)$, the corresponding perpendicular wavenumber resolutions are $k_{x,min} = k_{y,min} = 0.05\rho_{ti}^{-1}$, $k_{x,max} = k_{y,max} = 33.9\rho_{ti}^{-1}$ for the $T_e/T_i = 1$ case and $k_{x,max} = k_{y,max} = 16.95\rho_{ti}^{-1}$ for the $2 \le T_e/T_i$ cases. For single ion-scale turbulence simulations, we employed reduced perpendicular wavenumber space $k_{x,max} = 4\rho_{ti}^{-1}$, $k_{y,max} = 1\rho_{ti}^{-1}$, which well resolves large ion-scale micro-instabilities and associated turbulence but excludes small electron-scale dynamics. Further, for single electron-scale turbulence simulations, we employed smaller perpendicular box sizes $k_{x,min} = k_{y,min} = 0.5\rho_{ti}^{-1}$, which covers only electron-scale micro-instabilities but excludes ion-scale ones.

**Tracer particle analysis**. In Fig. 2, tracer particle trajectories are calculated to explain behaviour of resonant and off-resonant particles and to examine the effects of small electron-scale turbulence on the trajectories. Because of the low $\beta$ value, magnetic perturbations are neglected in diagnostics. The equation of motion of a gyrokinetic electron is solved up to the lowest order of $\rho_{te}/R_0$,

$$\frac{d\mathbf{x}}{dt} = v_\parallel \mathbf{b} + \mathbf{v}_{ed} + \mathbf{v}_E, \quad \frac{dv_\parallel}{dt} = -\frac{\mu \nabla_\parallel B_0}{m_e}, \quad \frac{d\mu}{dt} = 0, \quad (1)$$

where $\mathbf{B} = B_0\mathbf{b}$, $\mathbf{v}_{ed}$, and $\mathbf{v}_E$ are the equilibrium magnetic field, electron magnetic drift, and $E \times B$ drift velocities, respectively. Spatio-temporal data of gyrophase-averaged electrostatic potential from multi-scale turbulence simulations are used for evaluating the $E \times B$ drift. Since the pitch angle at the poloidal angle $\theta = 0$ is set as $0.45\pi$, the parallel velocity and magnetic moment for a resonant ($\nu = 2v_{te}$) particle are $v_\parallel = 0.31v_{te}$, $\mu = 2.38T_e/B_0$ and $\nu = 0.39v_{te}$, $\mu = 3.72T_e/B_0$ for an off-resonant ($\nu = 2.5v_{te}$) particle. Without $E \times B$ flows, the conservation of the canonical angular momentum ensures that there is no radial displacement of a particle. Even when $E \times B$ flows exist, there will still be no net radial transport if the electrostatic potential is time-independent because particles move only along the static electrostatic potential contour. Therefore, net radial displacement of a collisionless particle is induced by fluctuating $E \times B$ flows. For the analysis of turbulent transport, a statistical correlation between the fluctuating $E \times B$ flows and perturbations of the plasma distribution functions is necessary to be evaluated.

**Definition of the velocity–space-dependent turbulent energy flux**. Instead of calculating trajectories of large numbers of particles, we have solved the time evolution of plasma distribution functions. Then, the correlation between a turbulent radial $E \times B$ flow and a perturbed distribution function is obtained by taking the average in homogeneous directions $x$ and $y$ in a simulation box $L_x \times L_y$ and over a range of time $t_0 < t < t_0 + T$. When the particle kinetic energy is multiplied, the velocity–space-dependent turbulent energy flux in Fig. 3c is expressed as

$$Q_e^v(z, v_\parallel, \mu) = \left(\frac{m_e v_\parallel^2}{2} + \mu B_0\right) \int_{t_0}^{t_0+T} \frac{dt}{T} \int_0^{L_x} \frac{dx}{L_x} \int_0^{L_y} \frac{dy}{L_y} \mathbf{v}_E \cdot \nabla x f_e, \quad (2)$$

where we retain the poloidal angle $\theta = z$ dependence to analyse the trapped-passing boundary, which is regarded as a microscopic heat flux per unit of velocity–space volume[46]. Taking velocity–space and poloidal integrals, one obtains the time-averaged turbulent energy flux $Q_e = \langle \int dv^3\ Q_e^v \rangle_\theta$, where angle brackets $\langle \cdots \rangle_\theta$ denote the flux-surface average.

**Definition of the turbulent energy flux spectrum**. Because the turbulent transport is a convolution of a turbulent radial $E \times B$ flow and a perturbed distribution function, the time-averaged perpendicular wavenumber spectrum of the turbulent energy flux is given by

$$Q_{e\mathbf{k}} = \int_{t_0}^{t_0+T} \frac{dt}{T} \mathrm{Re}\left[\left\langle -\frac{\mathbf{b}\cdot\nabla x \times \nabla y}{B} ik_y \phi_\mathbf{k} p_{e\mathbf{k}}^* \right\rangle_\theta\right]. \quad (3)$$

The gyrophase-averaged perturbed electron pressure is denoted by $p_{e\mathbf{k}} = \int dv^3 (m_e v_\parallel^2/2 + \mu B_0) J_0(k_\perp \rho_e) f_{e\mathbf{k}}$ with the zeroth-order Bessel function $J_0$ and the perpendicular wavenumber $k_\perp$. The poloidal wavenumber spectra are calculated by taking the summation of $Q_{e\mathbf{k}}$ over $k_x\rho_{ti}$, $Q_{eky} = \Sigma_{kx} Q_{e\mathbf{k}}$. In Fig. 4, the plot is normalised to compare the simulations with different minimum wavenumber $\Delta k_y = k_{y,min}$ on an equal footing, $Q_e = \sum_{ky} Q_{eky} = \int dk_y (Q_{eky}/\Delta k_y)$.

**Definition of the non-linear triad transfer function**. The fluctuation intensity of distribution functions is measured by the perturbed entropy variable[47]. The entropy balance equation describes the entropy production due to transport fluxes

under thermodynamic gradient forces, which balances the collisional dissipation in a steady state[48]. In the perpendicular wavenumber space, the quadratic non-linearity of the $E \times B$ advection term characterises the triad interactions[35,49]. The gyrokinetic triad transfer function for the electron entropy balance is given by

$$J_{\mathbf{k}}^{\mathbf{p},\mathbf{q}}(t) = \delta_{\mathbf{k}+\mathbf{p}+\mathbf{q}=0} \frac{\mathbf{b} \cdot \mathbf{p} \times \mathbf{q}}{2B_0} \mathrm{Re}\left[\left\langle \int d v^3 \left(\chi_{\mathbf{p}} g_{e\mathbf{q}} - \chi_{\mathbf{q}} g_{e\mathbf{p}}\right) \frac{T_e g_{e\mathbf{k}}}{F_{e\mathrm{M}}} \right\rangle_\theta\right], \quad (4)$$

where $\chi_{\mathbf{k}} = \langle \phi_{\mathbf{k}} - v_\parallel A_{\parallel \mathbf{k}} \rangle$ and $g_{e\mathbf{k}} = f_{e\mathbf{k}} + e_e \langle \phi_{\mathbf{k}} \rangle F_{e\mathrm{M}}/T_e$ denote the gyrophase-averaged generalised potential and the non-adiabatic part of the perturbed electron distribution function for a mode $\mathbf{k}$. $F_{e\mathrm{M}}$ is the equilibrium Maxwell distribution. The triad transfer function $J_{\mathbf{k}}^{\mathbf{p},\mathbf{q}}$ describes the interaction of entropy variables among the three wavenumber modes $\{\mathbf{k}, \mathbf{p}, \mathbf{q}\}$ satisfying the non-linear coupling condition $\mathbf{k} + \mathbf{p} + \mathbf{q} = 0$. Namely, positive or negative $J_{\mathbf{k}}^{\mathbf{p},\mathbf{q}}$ means entropy gain or loss of mode $\mathbf{k}$ via the coupling with $\mathbf{p}$ and $\mathbf{q}$. The detailed balance relation $J_{\mathbf{k}}^{\mathbf{p},\mathbf{q}} + J_{\mathbf{p}}^{\mathbf{q},\mathbf{k}} + J_{\mathbf{q}}^{\mathbf{k},\mathbf{p}} = 0$ ensures entropy conservation.

To extract the large ion-scale and small electron-scale contributions separately, the subspace transfer function is defined as[36,50]

$$J_{\mathbf{k}}^{\Omega_s,\Omega_{s'}}(t) = \sum_{\mathbf{p} \in \Omega_s} \sum_{\mathbf{q} \in \Omega_{s'}} J_{\mathbf{k}}^{\mathbf{p},\mathbf{q}}, \quad (5)$$

where $\Omega_i$ and $\Omega_e$ are subspaces in wavenumber space corresponding to ion and electron scales. The subspace transfer function represents the energy gain or loss of the analysed mode via the coupling with subspaces. In Fig. 5, we split the wavenumber space into the ion-scale $\Omega_i = \{\mathbf{k} \,|\, {-}4 < k_x \rho_{ti} < 4, \, {-}1 < k_y \rho_{ti} < 1\}$ and electron-scale $\Omega_i = \{\mathbf{k} \,|\, \text{the others}\}$. Although there is the arbitrariness of the boundary choice between ion and electron scales, doubling the $|k_y \rho_{ti}|$ boundary from 1 to 2 gives no qualitative difference to the analysis because the peaks of the ion and electron scales are well separated as shown in Fig. 1. The subspace transfer also satisfies the symmetric property $J_{\mathbf{k}}^{\Omega p,\Omega q} = J_{\mathbf{k}}^{\Omega q,\Omega p}$, and the detailed balance among three subspaces $J_{\mathbf{k}\in\Omega k}^{\Omega p,\Omega q} + J_{\mathbf{k}\in\Omega p}^{\Omega q,\Omega k} + J_{\mathbf{k}\in\Omega q}^{\Omega k,\Omega p} = 0$.

## Data availability
The data depicted in the plots of this paper will be made available at the following https://github.com/smaeyama/maeyama_ncomm_2022 upon publication[51].

## Code availability
The GKV code is an open-source project available from GitHub: https://github.com/GKV-developers/gkvp. The version of the GKV code used in this paper is gkvp_f0.59. The source code with relevant physical/numerical parameter settings are available[51].

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

## Acknowledgements

One of the authors (S.M.) acknowledges helpful discussion with Dr. M. Honda. All the authors were supported by MEXT as 'Programme for Promoting Researches on the Supercomputer Fugaku' (Exploration of burning plasma confinement physics, JPMXP1020200103). S.M. was supported by JSPS KAKENHI Grant Number JP20K03892. S.M. and Y.A. were supported by 'Joint Usage/Research Centre for Inter-disciplinary Large-scale Information Infrastructures' and 'High-Performance Computing Infrastructure'. S.M. was supported by QST Research Collaboration for Fusion DEMO. S.M. and T.-H.W. acknowledge the NIFS Collaboration Research Programme (NIFS20KNST162, NIFS21KNST181) in Japan. Numerical analyses were performed on the Fugaku supercomputer at RIKEN R-CCS (Project ID: hp200127, hp210178), the JFRS-1 at Computational Simulation Centre of International Fusion Energy Research Centre (IFERC-CSC), and the plasma simulator at National Institute for Fusion Science.

## Author contributions

S.M. performed numerical simulations and analysed the data. T.-H.W. led the project of exploration of burning plasma confinement physics. M. Nakata, M. Nunami, and A.I. contributed to the physical interpretation. Y.A. contributed to the optimisation of the simulation code on the Fugaku supercomputer. All authors contributed to the development of the simulation code and the preparation of the manuscript.

## Competing interests

The authors declare no competing interests.
