## [Peer Review File · Nature Communications]

Multi-scale turbulence simulation suggesting improvement of electron heated plasma confinementReviewers' Comments:

Reviewer #1:

Remarks to the Author:

This is a very short report that details a novel observation of ion-scale TEM transport suppressed when resolving electron scale ETG fluctuations. This finding is contrary to the more common multiscale result that resolving short wavelength ETG modes increases both ion and electron flux.

There are numerous problems with the manuscript. Roughly in order of severity, these are

1. the numerical results are limited (1 ion-scale and 1 multi-scale case)
2. the plasma parameters are arguably unphysical ($T_e/T_i=3$ and $R/L_{ti} \sim 0$)
3. the theoretical explanation of stabilization is qualitative and unconvincing
4. large visualizations are presented that do not aid the analysis

Regarding 1, it can be argued that by modern standards, numerical gyrokinetic publications ought to include ensembles of cases showing curves of (say) flux versus a critical parameter like T_e/T_i or a/L_{ti} . To include only a single multiscale simulation seems inadequate. It might be justified for work that otherwise had a convincing array of ion or intermediate-scale results, but that is not the case for the present paper.

Regarding 2, the relevance of the chosen parameters to burning plasma simulation is dubious at best. In particular, the ratio $T_e/T_i=3$ is unphysically large even for a reactor due to the ion-electron energy exchange. ITER profiles modeling, for example, shows that even $T_e/T_i=2$ is unlikely. The ion temperature gradient 10x lower than the electron gradient is also an extreme assumption that makes the relevance to future burning plasma devices questionable.

Regarding 3, the time-dependent plots are confusing. In gyrokinetic theory, the time-dependence has limited meaning. Typically, only the time-average of a statistically steady state is meaningful. Early-time behaviour is transient.

Regarding 4, the colourful fluctuation visualizations do not support the physics claims of the paper and should probably be omitted in such a brief publication. Surely, the gyrokinetic equations can also be omitted in a paper of such brevity, and replaced by references to prior publications.

Although I find the prospect of "TEM stabilization by ETG" intriguing, I cannot accept this paper for publication based on the issues 1-4 above. Other issues of lower severity could be dealt with at that time.

I would be willing to review a substantially revised manuscript that address these concerns.

Reviewer #2:

Remarks to the Author:

Referee's Report on "Multi-scale turbulence simulation of ... " by S. Maeyama et al.

This manuscript reports on new findings which are interesting and potentially of high relevance to fusion. Simulation capabilities are quite impressive as well and probably no further simulation seems necessary before its publication.

However, analyses could have been more thorough and physics discussion should be strengthened considerably. Some remarks on the relevance to burning plasmas and on TEM are somewhat misleading. This paper needs to be reviewed again after a revision addressing the following points.

Interesting possibility has been made regarding the influence of ETG on trapped particles resonant with TEM as shown in Fig. 1. Initial difference between multiscale simulation results in red and TEM-only resonant case in black dashed line can be understood. However, using the word "precession resonance" for trapped particle-ETG interaction may not be appropriate here because the ETG wave period is presumably shorter than the trapped electron bounce time such that the bounce-averaged precession is not well-defined during ETG wave period. In that sense, it could be more important to discuss why red line for the radial displacement deviates secularly from black line starting at $t \approx 700$. In addition, why co-existence of ETG-eddies "means" that electron heat transport caused by TEMs is prevented by those as stated at the end of page 7? These are obviously correlated, but the logic behind the statement is not clear.

2. Mutual suppression of the large-scale turbulence by the short-scale turbulence and vice versa are interesting results. However, if one considers spectral transfer of energy due to nonlinear mode coupling, it sounds like both ion scale low-k mode and electron-scale high-k mode lose energy as a result of nonlinear interaction. Then, where does that energy go? Is there any signature of electron heating due to trapped electron Compton scattering (nonlinear Landau damping), for instance?

3. In Eq. (4), why is there no E_{\parallel} in the $(dv_{\parallel})/dt$ equation? Is it a typographical error?

4. "Burning plasma relevance" of this work has been over-emphasized on page 9 and page 18.

a) Why are fusion-borne alpha particles not considered in simulations while He ash is kept? Perhaps, the results from [S.M.Yang et al., PoP 25, 122305 (2018)] reporting low level of α -particle transport from TEM could provide a partial justification.

b) Separately reporting $Q_D + Q_T$ and Q_{He} values can be useful since one is harmful and the other is beneficial for fusion.

c) While this is an impressive simulation effort, some simplification of parameters is inevitable and understandable. It is better to state that choices of parameters $T_e = 3T_i$, $R_0/L_{Ti} = 1$, etc are made for a study focusing on TEM-ETG interaction. What is T_e value in keV?

d) It is not obvious why electromagnetic simulation is performed for a very low value of $\beta = 0.05\%$. If there is any considerable electromagnetic effect observed, state it.

5. Presentations :

a) Refs 9-12 mostly deal with ETG-driven transport level estimation from simulations. Experimental observation of ETG-scale turbulence is at least equally important. For instance, [E. Mazzucato et al., PRL 101, 075001 (2008)] should be cited.

b) Discussion regarding T_e/T_i dependence at the end of page 3 is misleading in the context of this paper. Then, one might expect ITG rather than TEM should have been considered for $T_e = 3T_i$ for a realistic value of R_0/L_{Ti} .

c) In future devices, disparity between p_i and system size would be even more pronounced so that meso-scale turbulence and its interaction with ion scale turbulence [eg., T.S.Hahm and P.H.Diamond JKPS 73, 747 (2018)] would become another relevant example of multi-scale interaction. This needs to be stated in the introduction.

d) It would have made more sense to choose k_y -modes at the peak of nonlinearly saturated spectra, rather than most linearly unstable ones for the analyses reported on page 5 and beyond. Present k -spectra at nonlinear saturation if possible by any means.

e) The reason stated for TEM consideration in the middle of page 5 is misleading. There have been considerable nonlinear analytic theory efforts addressing TEM's relevance to experiments [eg., P.L.Similon and P.H.Diamond, PF 27, 916 (1984); T.S. Hahm and W.M.Tang, PFB 3, 989 (1991)] before recent gyrokinetic simulations such as Refs 26 and 27. Aforementioned papers also discuss the role of Compton scattering in relation to the item 2 in this report.

Reply to the Reviewer #1

We would like to express our deep gratitude to the reviewer for highlighting the inadequacies in our manuscript and providing suggestions for improvement. We have revised the manuscript following the reviewer's comments (in italic bold font below).

There are numerous problems with the manuscript. Roughly in order of severity, these are

- 1. the numerical results are limited (1 ion-scale and 1 multi-scale case)*
- 2. the plasma parameters are arguably unphysical ($T_e/T_i=3$ and $R/L_i \sim 0$)*
- 3. the theoretical explanation of stabilization is qualitative and unconvincing*
- 4. large visualizations are presented that do not aid the analysis*

Regarding 1, it can be argued that by modern standards, numerical gyrokinetic publications ought to include ensembles of cases showing curves of (say) flux versus a critical parameter like T_e/T_i or a/L_i . To include only a single multiscale simulation seems inadequate. It might be justified for work that otherwise had a convincing array of ion or intermediate-scale results, but that is not the case for the present paper.

As following the reviewer's suggestion, we have increased the number of simulations to four multi-scale studies and associated ion-scale and electron-scale cases. The maximum resolution in a case $T_e/T_i = 1$ was increased fourfold, $(N_x, N_y, N_z, N_{v||}, N_\mu) = (2048, 2048, 40, 64, 16)$ to resolve high-wavenumber ETG modes. Total computing time requirements were 2 M node-hour (98 M core-hour) on the Fugaku supercomputer. These additional simulation results are summarised in curves of electron heat flux versus temperature ratio T_e/T_i in Fig. 6 in the revised manuscript.

Regarding 2, the relevance of the chosen parameters to burning plasma simulation is dubious at best. In particular, the ratio $T_e/T_i=3$ is unphysically large even for a reactor due to the ion-electron energy exchange. ITER profiles modeling, for example, shows that even $T_e/T_i=2$ is unlikely. The ion temperature gradient 10x lower than the electron gradient is also an extreme assumption that makes the relevance to future burning plasma devices questionable.

We agree that the relevance of our simulations for future burning plasma needs to be carefully discussed.

We have added the parameter scan over the range of $1 \leq T_e/T_i \leq 4$ as Fig. 6 in the revised manuscript. The ratio $T_e/T_i = 3$ is the case where the linear ETG growth rates are near marginal (ETGs are stable at $T_e/T_i = 4$). As observed, ETGs are strongly unstable at $T_e/T_i \leq 2$ and have a significant contribution to electron heat flux. Therefore, our simulations showed that ETGs could contribute to electron turbulent transport in a much wider range of $1 < T_e/T_i$. We have added this sentence to the revised manuscript, “From the ITER profile modelling and a reference value of a high T_e discharge of the DIII-D tokamak [Grierson, Phys. Plasmas (2019)], a reactor-relevant temperature ratio is considered around $1 < T_e/T_i < 2$ due to the ion–electron energy exchange. Our survey of dependence of Q_e on T_e/T_i covers this range. An important consequence is that the ETG contribution survives in a wider parameter range $1 < T_e/T_i < 3$, although it may have been considered that ETG could have non-negligible contribution when $T_e \sim T_i$ and sufficiently large electron temperature [Mariani, Nucl. Fusion (2021)]. Additionally, the multi-scale turbulence simulation shows the existence of an appropriate T_e/T_i range where the cross-scale interactions suppress turbulent transport. The suppression of near-marginal TEM by ETG turbulence leads the upshift of the critical temperature ratio for increase of TEM-dominated energy flux (from $T_e/T_i \sim 2$ in the ion-scale simulation to $T_e/T_i \sim 3$ in the multi-scale simulation), analogous to the Dimits shift where zonal flows suppress near marginal ITG [Dimits, Phys. Plasmas (2000)].”

When we employ a large ion temperature gradient, ITG modes become unstable. For the case of $T_e/T_i = 1$ and $R_0/L_{Ti} = R_0/L_{Te} = 6.97$, we have already reported the detailed mechanisms of multi-scale interactions between ITG/ETG turbulence in our previous publications. To avoid repeating the same discussions as previous publications, we have added the following sentences in the revised manuscript: “When the ion temperature gradient increases due to the energy exchange, the ITG modes will be destabilised. For the multi-scale ITG/ETG turbulence at $T_e = T_i$ and $R_0/L_{Te} = R_0/L_{Ti}$, see our previous publications [Maeyama, PRL (2015); Nucl. Fusion (2017)], where detailed mechanisms of cross-scale interactions between ITG and ETG modes were reported. In this study, we focus on the parameter regime at $T_e > T_i$ and $R_0/L_{Te} > R_0/L_{Ti}$ which has not yet been analysed in existing devices” and “These parameters are also related to the electron heated plasma at the PFPO-1 phase in the ITER research plan which is an important step for early success of ITER. The prediction of the integrated modeling of PFPO-1 phase *H*-mode plasma reported high central temperature ratio

$T_e/T_i > 3$ and $R_0/L_{Te} > R_0/L_{Ti}$ [Loarte, Nucl. Fusion (2021)].” Therefore, our survey at $T_e/T_i = 3$ is not unphysical.

We have carefully revised the manuscript about the relevance of our simulation for burning plasma. Especially, the title is modified to more precise one, “Multi-scale turbulence simulation suggesting improvement of electron heated plasma confinement”.

Regarding 3, the time-dependent plots are confusing. In gyrokinetic theory, the time-dependence has limited meaning. Typically, only the time-average of a statistically steady state is meaningful. Early-time behaviour is transient.

Following the suggestion, Fig. 3 in the previous manuscript has been replaced with the time-averaged poloidal wavenumber spectra of electron energy flux. (Fig. 4 in the revised manuscript). Figure 4 in the previous manuscript has also been replaced with the time-averaged perpendicular wavenumber spectra of non-linear electron entropy transfer (Fig. 5 in the revised manuscript).

Regarding 4, the colourful fluctuation visualizations do not support the physics claims of the paper and should probably be omitted in such a brief publication. Surely, the gyrokinetic equations can also be omitted in a paper of such brevity, and replaced by references to prior publications.

Following the suggestion, we have removed the visualisation of Extended Data Fig. 2 in the manuscript. Gyrokinetic equations [Eqs. (1)–(3)] in the manuscript have also been removed.

All modifications related to the comments of reviewer #1 are highlighted in red in the revised manuscript.

Reply to the Reviewer #2

We would like to express our deep gratitude to the reviewer for their careful reading and constructive discussions. We have revised the manuscript following the reviewer's comments (in italic bold font below).

1. Interesting possibility has been made regarding the influence of ETG on trapped particles resonant with TEM as shown in Fig. 1. Initial difference between multiscale simulation results in red and TEM-only resonant case in black dashed line can be understood. However, using the word “precession resonance” for trapped particle-ETG interaction may not be appropriate here because the ETG wave period is presumably shorter than the trapped electron bounce time such that the bounce-averaged precession is not well-defined during ETG wave period. In that sense, it could be more important to discuss why red line for the radial displacement deviates secularly from black line starting at $t \approx 700$.

We agree that we need to use the word “precession resonance” carefully because of the time-scale separation between TEM (or typical bounce period) and ETG modes. As explained in the manuscript, the wave phase velocity is close for TEM and ETG modes [$(\omega_r/k_{\text{tor}})_{\text{TEM}} = 42.7 v_{\text{ti}}\rho_{\text{ti}}/R_0$ and $(\omega_r/k_{\text{tor}})_{\text{ETG}} = 45.1 v_{\text{ti}}\rho_{\text{ti}}/R_0$], whereas the wave frequency is quite different ($\omega_{\text{r,ETG}} \gg \omega_{\text{r,TEM}}$) due to differences in the wavenumber ($k_{\text{tor,ETG}} \gg k_{\text{tor,TEM}}$). Therefore, TEM-relevant trapped particles [moving toroidally by the magnetic drift, $v_{\text{pre}} = qv_{\text{dy}}/\varepsilon_r \sim m_e v^2 q / (2erB_0)$, where $v_{\text{dy}} = \mathbf{v}_d \cdot \nabla y$ is the contravariant component of magnetic drift velocity in the field-aligned coordinates (x, y, z)] interact with ETG modes propagating the same phase velocity ($v_{\text{pre}} \sim \omega_{\text{r,TEM}}/k_{\text{tor,TEM}} \sim \omega_{\text{r,ETG}}/k_{\text{tor,ETG}}$). In this process, a bounce average is not a requirement. In the revised manuscript, we modified the expressions:

- Line 121 (page 6) and line 141 (page 7): from “during the precession drift” to “during the toroidal drift motion”.
- Line 123 (page 6): from “over a toroidal precession path length” to “over a toroidal path length”. The same modification has been applied to the x-axis label in Fig. 2.
- Line 269 (page 15): from “disturbance of precession drift resonance of trapped electrons by ETG in this study” to “disturbance of drift resonance between trapped electrons and TEM by ETG in this study”

Regarding the tracer particle simulation, the multi-scale and without electron-scale

cases are radially advected outward using a large-scale TEM eddy until $yq/(\varepsilon_r \rho_{ti}) \sim 400$. In the case without electron-scale fluctuations, the particle moves with the same eddy even after $yq/(\varepsilon_r \rho_{ti}) > 400$. On the other hand, in the multi-scale case, the particle trajectory is stirred by electron-scale fluctuations (seen as spikes in E_y) and moved on a different large-scale TEM eddy at $yq/(\varepsilon_r \rho_{ti}) \sim 700$, which is the reason for the occurrence of a secular displacement. The above explanation has been added to the revised manuscript. The secular inward/outward displacement depends on the initial position/velocity of tracer particles and turbulent flows. Therefore, net effects on turbulent transport should be quantitatively evaluated using statistical treatment, as discussed in the manuscript.

In addition, why co-existence of ETG-eddies “means” that electron heat transport caused by TEMs is prevented by those as stated at the end of page 7? These are obviously correlated, but the logic behind the statement is not clear.

As mentioned by the reviewer, the co-existence of ETG and TEM does not directly mean the prevention of heat transport in TEM, which was observed by comparing multi-scale and ion-scale simulations in Figs. 4 and 6 and by the non-linear transfer analysis in Fig. 5. Since the previous sentence was misleading, we modified the expression in the revised manuscript.

2. Mutual suppression of the large-scale turbulence by the short-scale turbulence and vice versa are interesting results. However, if one considers spectral transfer of energy due to nonlinear mode coupling, it sounds like both ion scale low-k mode and electron-scale high-k mode lose energy as a result of nonlinear interaction. Then, where does that energy go? Is there any signature of electron heating due to trapped electron Compton scattering (nonlinear Landau damping), for instance?

We have investigated the spectral transfer of energy (more precisely, non-linear triad transfer of gyrokinetic entropy). Entropy conservation (called the detailed balance $J_k^{p,q} + J_p^{q,k} + J_q^{k,p} = 0$) is confirmed theoretically and numerically. In the revised manuscript, Fig. 5 plots the perpendicular wavenumber spectra of the non-linear entropy transfer, where contributions via coupling with ion scales (denoted by Ω_i) and electron scales (denoted by Ω_e) are summarised by means of the subspace transfer analysis technique [Maeyama, Nucl. Fusion (2017)]. It is shown that electron-scale, electron-scale coupling has a negative contribution on TEM ($J_k^{\Omega_e, \Omega_e} < 0$ at $k_y \rho_{ti} < 0.5$),

and similarly, ion-scale, electron-scale coupling has negative contribution on ETG ($J_k^{\Omega_i, \Omega_e} < 0$ at $k_y \rho_{ti} \sim 4$ and $k_x \rho_{ti} \sim 0$, characterized by a peak of energy spectrum at high wavenumber, which is close to linearly unstable ETG modes). This observation directly supports mutual suppression between ETG and TEM turbulence. In addition, from the viewpoint of the detailed balance in the subspace transfer analysis $J_{k \in \Omega_i}^{\Omega_e, \Omega_e} + 2 J_{k \in \Omega_e}^{\Omega_i, \Omega_e} = 0$, the suppression of TEM by ETG ($J_k^{\Omega_e, \Omega_e} < 0$ at $k_y \rho_{ti} < 0.5$) means the entropy transfer from low- k_y TEM mode to high- k_y modes ($2 J_{k \in \Omega_e}^{\Omega_i, \Omega_e} = -J_{k \in \Omega_i}^{\Omega_e, \Omega_e} > 0$). As plotted, net entropy gain is observed at high- k_y and finite k_x modes ($J_k^{\Omega_i, \Omega_e} > 0$ at $k_y \rho_{ti} \sim 4$ and $k_x \rho_{ti} > 1$), but not at a ETG peak ($k_y \rho_{ti} \sim 4$ and $k_x \rho_{ti} \sim 0$). These high- k_\perp modes creates higher k_\perp fluctuations ($J_k^{\Omega_e, \Omega_e} > 0$ at higher $k_y \rho_{ti} > 7$, not shown), which are eventually damped by collisional dissipation. The above discussion is added in page 12.

Regarding trapped electron Compton scattering, it may be responsible for the saturation of TEM turbulence as discussed in the the suggested references [Similon, PF (1984); Hahm, PFB (1991)]. We are not sure whether it can describe the present cross-scale interactions between TEM and ETG modes, where the coupling among a TEM and two higher k_y modes are responsible for $J_k^{\Omega_e, \Omega_e} < 0$ at $k_y \rho_{ti} < 0.5$.

3. In Eq. (4), why is there no E_\parallel in the (dv_\parallel/dt) equation? Is it a typographical error?

For the tracer particle trajectory analysis, the parallel electric force $eE_\parallel/m_e \sim (\rho/R_0)k_\parallel T/m_e$, which is one order smaller than the mirror force, cannot appear in the lowest order equation. Of course, in the gyrokinetic Vlasov simulation, this term appears in the equilibrium contribution $-(eE_\parallel/m_e)\partial F_{eM}/\partial v_\parallel$ [but not in the parallel nonlinearity for the perturbed distribution function $-(eE_\parallel/m_e)\partial f_\delta/\partial v_\parallel$], as being consistent with the conventional gyrokinetic ordering.

4. “Burning plasma relevance” of this work has been over-emphasized on page 9 and page 18.

We agree that the relevance of our simulations for future burning plasma needs to be carefully discussed. We have carefully revised the manuscript and modified the title to more precise one, “Multi-scale turbulence simulation suggesting improvement of electron heated plasma confinement”.

a) Why are fusion-borne alpha particles not considered in simulations while He ash is kept? Perhaps, the results from [S.M.Yang et al., PoP 25, 122305 (2018)] reporting low level of α -particle transport from TEM could provide a partial justification.

In the revised manuscript, we mentioned the limitations of our study and future work. “This work is limited to simulations with electrons, deuterium and tritium fuels and helium ash, excluding energetic alpha particle dynamics. It has been reported that TEM drives low-level transport of alpha particles [Yang, Phys. Plasmas (2018)]. The resonant interactions between the energetic particle and micro-instability will be significant for ITG modes [Di Siena, PRL (2020)], while not for TEM and ETG modes because the propagation directions are opposite [Mazzi, Nucl. Fusion (2020); Hussain, Plasma Phys. Control. Fusion (2021)]. Therefore, interactions between energetic particles and TEM/ETG turbulence are expected to be weak. When we consider interplay from electron to ion-scale turbulence and macroscopic fluctuations, magnetohydrodynamic instability driven by energetic particles will be a candidate for the third player [Ishizawa, Nucl. Fusion (2021)], which is a challenging subject theoretically and numerically.”

b) Separately reporting Q_D+Q_T and Q_{He} values can be useful since one is harmful and the other is beneficial for fusion.

We have separated these values in the revised manuscript.

c) While this is an impressive simulation effort, some simplification of parameters is inevitable and understandable. It is better to state that choices of parameters $T_e=3T_i$, $R_0/L_{Ti}=1$, etc are made for a study focussing on TEM-ETG interaction. What is T_e value in keV?

We have added the parameter scan over the range of $1 \leq T_e/T_i \leq 4$ as Fig. 6 in the revised manuscript, similar to the answer to a comment from reviewer #1. We have added this sentence to the revised manuscript, “From the ITER profile modelling and a reference of a high T_e discharge of DIII-D #173147 [Grierson, Phys. Plasmas (2019)], a reactor-relevant temperature ratio is considered around $1 < T_e/T_i < 2$ due to the ion–electron energy exchange. Our survey of dependence of Q_e on T_e/T_i well covers this range and shows that ETG contributions survive in a wide parameter range

of $1 < T_e/T_i < 3$.”

We have also added the following sentences focussing on TEM/ETG turbulence: “When the ITG increases due to the energy exchange, the ITG modes will be destabilised. For the multi-scale ITG/ETG turbulence at the $T_e = T_i$ and $R_0/L_{Te} = R_0/L_{Ti}$ case, see our previous publications [Maeyama, PRL(2015); Nucl. Fusion (2017)], where detailed mechanisms of cross-scale interactions between ITG and ETG modes were reported. In this paper, we focus on the parameter regime at $T_e > T_i$ and $R_0/L_{Te} > R_0/L_{Ti}$ which has not yet been well analysed in existing devices” and “These parameters are also related to the electron heated plasma at the PFPO-1 phase in the ITER research plan which is an important step for early success of ITER. The prediction of the integrated modeling of PFPO-1 phase H-mode plasma reported high central temperature ratio $T_e/T_i > 3$ and $R_0/L_{Te} > R_0/L_{Ti}$ [Loarte, Nucl. Fusion (2021)].”

In the local gyrokinetic simulation, the temperature is not determined in keV. All required normalised parameter is listed in the manuscript.

d) It is not obvious why electromagnetic simulation is performed for a very low value of $\beta=0.05\%$. If there is any considerable electromagnetic effect observed, state it.

Turbulence dynamics are almost electrostatic; there is no considerable electromagnetic effect. The reason why we carried out electromagnetic simulation is that we worried about a numerical time-step-size restriction by the so-called high-frequency mode ω_H in the electrostatic limit [Lin, PoP (2001)] because the GKV code employs the explicit time integration for collisionless dynamics.

5. Presentations :

a) Refs 9-12 mostly deal with ETG-driven transport level estimation from simulations. Experimental observation of ETG-scale turbulence is at least equally important. For instance, [E. Mazzucato et al., PRL 101, 075001 (2008)] should be cited.

We have cited the literature in the revised manuscript.

b) Discussion regarding T_e/T_i dependence at the end of page 3 is misleading in

the context of this paper. Then, one might expect ITG rather than TEM should have been considered for $T_e=3T_i$ for a realistic value of $R_{0L_{Ti}}$.

What we intended is that the ion-scale instabilities (ITG or temperature gradient driven TEM) tend to be destabilised, whereas electron-scale instabilities (ETG) tend to be stabilised. It depends on other parameters whether TEM or ITG dominates at ion scale.

c) In future devices, disparity between ρ_i and system size would be even more pronounced so that meso-scale turbulence and its interaction with ion scale turbulence [eg., T.S.Hahm and P.H.Diamond JKPS 73, 747 (2018)] would become another relevant example of multi-scale interaction. This needs to be stated in the introduction.

We have cited the literature and mentioned the above class of multi-scale interactions in the revised manuscript.

d) It would have made more sense to choose k_y -modes at the peak of nonlinearly saturated spectra, rather than most linearly unstable ones for the analyses reported on page 5 and beyond. Present k -spectra at nonlinear saturation if possible by any means.

We have added k_y spectra as Fig. 4 in the revised manuscript.

e) The reason stated for TEM consideration in the middle of page 5 is misleading. There have been considerable nonlinear analytic theory efforts addressing TEM's relevance to experiments [eg., P.L.Similon and P.H.Diamond, PF 27, 916 (1984); T.S. Hahm and W.M.Tang, PFB 3, 989 (1991)] before recent gyrokinetic simulations such as Refs 26 and 27. Aforementioned papers also discuss the role of Compton scattering in relation to the item 2 in this report.

Thank you for pointing out important literature about TEM. We have cited them in the revised manuscript.

All modifications related to the comments of reviewer #2 are highlighted in blue in the revised manuscript.

Reviewers' Comments:

Reviewer #1:

Remarks to the Author:

SUMMARY:

I recommend this article for publication, assuming that the authors will make two modest figure additions.

REVIEW:

I thank the authors for responding thoughtfully to all four problems that were listed in my original review. Most importantly, in problem 1, I suggested that additional simulations should be carried out. In return, the authors ran and summarized an additional three simulations at $T_e/T_i=(1.0,2.0,4.0)$ to complement the original simulation at $T_e/T_i=3.0$. This clearly took a significant amount of computational effort and I believe the manuscript has benefited strongly from this improvement. The authors also took care to remedy the issues raised in problems 2-4.

Because of the authors' satisfactory responses to all problems, I recommend the article for publication but STRONGLY suggest that the following additions be made:

1. Please include linear results (as in Fig 1) for all three new simulations ($T_e/T_i=1,2,4$)
2. Please include nonlinear spectra (as in Fig 4) for all three new simulations

Finally, as a tentative third suggestion:

3. Consider using a vertical log scale for Fig 6.

Reviewer #2:

Remarks to the Author:

This version of manuscript is significantly improved through a revision addressing my previous suggestions. I recommend its publication in Nature Communications without a further review. Authors may want to add a few words on page 14 for a clarification for non-expert readers.

'...the Dimits upshift of critical ion temperature gradient where ...'

Reply to the Reviewer #1

We would like to express our deep gratitude to the reviewer for recommendation for publication. We have revised the manuscript following the reviewer's comments (in italic bold font below).

Because of the authors' satisfactory responses to all problems, I recommend the article for publication but STRONGLY suggest that the following additions be made:

- 1. Please include linear results (as in Fig 1) for all three new simulations ($T_e/T_i=1,2,4$)*
- 2. Please include nonlinear spectra (as in Fig 4) for all three new simulations*

These figures are provided in the supplementary information.

Finally, as a tentative third suggestion:

- 3. Consider using a vertical log scale for Fig 6.*

A vertical log scale is used for Fig. 6. Additionally, we have found previous low transport levels at $T_e/T_i=1$ cases vary in time. Through changing the resolution and box size and evaluating long time simulations, we have replaced the value at $T_e/T_i = 1$ cases, which is consistent with the spectra shown in the Supplementary Fig. 2.

Reply to the Reviewer #2

We would like to express our deep gratitude to the reviewer for recommendation for publication. We have revised the manuscript following the reviewer's comments (in italic bold font below).

Authors may want to add a few words on page 14 for a clarification for non-expert readers.

'...the Dimits upshift of critical ion temperature gradient where ...'

We have added the above words for explanation.

List of revisions

- Supplementary information file is provided.
- Page 13, line 238: The following sentence is added, “Corresponding linear dispersion relations (as in Fig. 1) and poloidal wavenumber spectra of electron energy flux (as in Fig. 4) for each T_e/T_i are respectively shown in Supplementary Figs. 1 and 2.”
- Page 14, line 256: We have modified the sentence as “the Dimits upshift of critical ion temperature gradient where ...”
- Page 14, Fig. 6: The vertical axis is changed to log scale. The values at $T_e/T_i = 1$ cases are modified to be consistent with the Supplementary Fig. 2.
- Page 15, line 263: The following sentence is added in the figure caption, “Note that an electron-scale simulation at $T_e/T_i = 4$ and ion-scale simulations at $T_e/T_i = 1$ and 2 contains no unstable modes (see Supplementary Fig. 1) and decays in time, and therefore cannot define finite amplitudes. The corresponding points are plotted on x-axis. A large standard deviation bar for the case of the electron-scale simulation at $T_e/T_i = 1$ (268 QgB) is omitted for visibility.”
- Page 30 line 604: The following sentence is added. “Supplementary information The online version contains supplementary material available.”

All modifications in the main manuscript is highlighted in red color.

Reviewers' Comments:

Reviewer #1:

Remarks to the Author:

I recommend this article for publication.